# SARS-CoV-2 Is Persistent in Placenta and Causes Macroscopic, Histopathological, and Ultrastructural Changes

**DOI:** 10.3390/v14091885

**Published:** 2022-08-26

**Authors:** André Luiz N. Parcial, Natália Gedeão Salomão, Elyzabeth Avvad Portari, Laíza Vianna Arruda, Jorge José de Carvalho, Herbert Leonel de Matos Guedes, Thayana Camara Conde, Maria Elizabeth Moreira, Marcelo Meuser Batista, Marciano Viana Paes, Kíssila Rabelo, Adriano Gomes-Silva

**Affiliations:** 1Interdisciplinary Laboratory of Medical Research, Instituto Oswaldo Cruz, Oswaldo Cruz Foundation, Rio de Janeiro 21040900, Brazil; 2Pathological Anatomy, Fernandes Figueira Institute, Oswaldo Cruz Foundation, Rio de Janeiro 22250020, Brazil; 3Laboratory of Ultrastructure and Tissue Biology, Rio de Janeiro State University, Rio de Janeiro 20551030, Brazil; 4Laranjeiras Health House, Rio de Janeiro 22240005, Brazil; 5Mycobacteriosis Clinical Research Laboratory, National Institute of Infectious Diseases Evandro Chagas, Oswaldo Cruz Foundation, Rio de Janeiro 21040900, Brazil

**Keywords:** COVID-19, pregnant women, viral particles, pathogenesis, inflammation

## Abstract

SARS-CoV-2 is a virus that belongs to the *Betacoronavirus* genus of the *Coronaviridae* family. Other coronaviruses, such as SARS-CoV and MERS-CoV, were associated with complications in pregnant women. Therefore, this study aimed to report the clinical history of five pregnant women infected with SARS-CoV-2 (four symptomatic and one asymptomatic who gave birth to a stillborn child) during the COVID-19 pandemic. They gave birth between August 2020 to January 2021, a period in which there was still no vaccination for COVID-19 in Brazil. In addition, their placental alterations were later investigated, focusing on macroscopic, histopathological, and ultrastructural aspects compared to a prepandemic sample. Three of five placentas presented SARS-CoV-2 RNA detected by RT-PCRq at least two to twenty weeks after primary pregnancy infection symptoms, and SARS-CoV-2 spike protein was detected in all placentas by immunoperoxidase assay. The macroscopic evaluation of the placentas presented congested vascular trunks, massive deposition of fibrin, areas of infarctions, and calcifications. Histopathological analysis showed fibrin deposition, inflammatory infiltrate, necrosis, and blood vessel thrombosis. Ultrastructural aspects of the infected placentas showed a similar pattern of alterations between the samples, with predominant characteristics of apoptosis and detection of virus-like particles. These findings contribute to a better understanding of the consequences of SARS-CoV-2 infection in placental tissue, vertical transmission.

## 1. Introduction

Severe acute respiratory syndrome, which occurred in 2002–2003, had its etiological agent identified as the severe acute respiratory syndrome coronavirus (SARS-CoV) [1] of the genus *Betacoronavirus* and of the *Coronaviridae* family [2]. In late 2019, a new severe acute respiratory syndrome coronavirus (SARS-CoV-2) emerged in China and has caused a worldwide pandemic, known as coronavirus disease 2019 (COVID-19) [3]. As of April 2022, SARS-CoV-2 has caused 494 million infections and 6.16 million deaths worldwide. The disease caused a great impact in Brazil, with more than 658,000 deaths until April 2022 (Brazil Health Ministry).

Coronaviruses are enveloped viruses with single-stranded RNA genomes, most of which encode 16 nonstructural proteins (nsp) and 9 accessory proteins. The rest of the genome encodes the so-called structural proteins called spike (S), envelope (E), membrane (M), and nucleocapsid (N) [4]. These genomic elements are shared by other coronaviruses [2]. However, SARS-CoV encodes specific accessory proteins, such as p3a and p3b, which are responsible for high virulence and exhibit functions between virus–host interactions during in vivo coronavirus infection [2,5,6,7].

The entry of SARS-CoV-2 into human host cells is determined by the interaction between the S viral protein receptor-binding domain (RBD) and the angiotensin-converting enzyme 2 (ACE2) cell receptor, and this mechanism is similar to that observed for SARS-CoV in that it shares significant homology in the RBD of the S protein [8]. The S protein of SARS-CoV-2 binds to the ACE2 receptor with greater affinity than SARS-CoV [9]. In addition to ACE2, the cleavage of the S protein by TMPRSS2 is indispensable for the entry of SARS-CoV-2 into target cells and for its successful spread [10,11]. Moreover, the confirmation of colocalization of the CD147 coreceptor with the spike protein binding of SARS-CoV-2 in gastrointestinal and lung tissues, as well as the reduction of infection in intestinal epithelial cells after neutralization of this coreceptor, suggests CD147 as a possible key molecule for the viral susceptibility of some tissues [12,13].

Primary viral replication is presumed to occur in the upper respiratory tract mucosal epithelium (nasal cavity and pharynx), with greater multiplication in the lower respiratory tract and gastrointestinal mucosa [14]. Moriyama and collaborators [15] suggest that SARS-CoV-2 can be transmitted to humans by air transmission, through direct, indirect, or close contact with infected people through infected secretions, such as short-range respiratory droplets, long-range aerosols, and fomites (contact with contaminated objects and surfaces).

The coexpression of ACE2 and TMPRSS2 in placental tissue has been observed in villous cytotrophoblast (CTB), syncytiotrophoblast (SCT), and extravillous trophoblast (EVT) cells of the maternal–fetal interface. Although there is expression of CD147 in trophoblast cells, these molecules are more present on the basal side of the cells. In other words, there is a possibility of coreceptor support, but this would probably occur in the case of a rupture of the placental barrier [16]. In contrast, the expression level of ACE2/TMPRSS2 may increase at the maternal–fetal interface along with the advancement of pregnancy [17]. Thus, several studies demonstrate the possibility of vertical transmission [18,19,20,21,22,23,24,25].

Viral infections have been linked to an increased risk of morbidity in pregnant women with infections by different types of coronaviruses (SARS-CoV, MERS-CoV). Miscarriage, premature birth, intrauterine growth retardation, premature rupture of membranes, fetal/neonatal death, and maternal death are examples of obstetric complications reported in these patients [26,27]. Although rare, vertical transmission of SARS-CoV-2 via the transplacental route has been reported [28,29].

In general, placentas in cases of congenital virus infection reveal hematogenous placentitis characterized by villitis, which can vary in extension and intensity, configuring granulomatous villositis and even microabscesses. In rare cases, it is possible to identify evidence of viral particles [30]. Some viruses, however, can cross the placental barrier, causing more villous stromal alterations (such as delayed villous maturation and Hofbauer cell hyperplasia) than inflammatory alterations (focal and/or mild multifocal villositis), as observed in cases of congenital Zika syndrome [31,32,33,34,35].

Morphological and molecular studies in the placentas of pregnant women infected by SARS-CoV-2 are scarce and inconclusive, and further studies are needed to elucidate the mechanism of this infection in the maternal–fetal binomial and a possible protective role of the placenta. Therefore, in this study, we investigated the presence of genome, proteins, and viral particles of SARS-CoV-2 in the placental tissue of infected pregnant women and observed the macroscopic, histopathological, and ultrastructural changes in these placentas.

## 2. Materials and Methods

### 2.1. Sample Collection and Storage

Placental samples were collected from five pregnant women, four symptomatic and one asymptomatic, during the COVID-19 pandemic, between August 2020 to January 2021, when there was still no vaccination in Brazil. All placental samples (control and infected) were from the Perinatal Laranjeiras e Barra or Fernandes Figueira Institute hospitals in Rio de Janeiro. The control placenta of a 33-year-old woman with no pre-existing diseases or during pregnancy was collected before the period of the COVID-19 epidemic in Brazil. These placentas were frozen or fixed in 10% buffered formalin or 2.5% glutaraldehyde.

### 2.2. Molecular Diagnosis

Placental fragments were collected in 1 mL of Trizol reagent and placed in liquid nitrogen until tissue processing. Tissue samples were subjected to 4 cycles of standardized mechanical dissociation (6 m/s, 30 s) using the L-Beader system (Loccus, São Paulo, SP, Brazil). After that, the material was centrifuged (460× *g*, 2 min) and the supernatant collected. A 5 μL volume of placental macerate in Trizol was applied in the semi-automated BDmax (BD) total nucleic acid extraction system and One-Step Real-Time RT-PCR. In summary, the BDmax system used the magnetic method of total DNA and RNA extraction in a standardized way, resulting in 25 μL of final volume. Immediately after total extraction, 12.5 μL of the extracted material was used by the system to elute the lyophilized BDmax master mix. This mixture was added to 12.5 μL of 2x concentrated primers and probe solution, finishing with 25 μL of solution for PCR reaction prepared by the BDmax system. Finally, the equipment applied 12.5 μL of the PCR solution in BDmax microplates, and each well used was sealed by heating. After that, the equipment started reverse transcription followed by real-time PCR for the E gene of SARS-CoV-2 according to the amplification conditions of the Berlin protocol using the following primer sequences and probe: E_Sarbeco_F ACAGGTACGTTAATAGTTAATAGCGT (400 nM), E_Sarbeco_R ATATTGCAGCAGTACGCACACA (400 nM) and E_Sarbeco_P1 FAM-ACACTAGCCATCCTTACTGCGCTTCG-BBQ (200 nM). In addition, a patented synthetic RNA was also used as an internal control for extraction and amplification in all reactions performed on the BDmax system.

### 2.3. Histopathological Analysis

Samples fixed in 10% buffered formalin were processed in increasing baths of ethanol (70, 90, and 100%), cleared in two xylol baths, infiltrated in paraffin for half an hour each, and finally embedded in paraffin. From the paraffin blocks with the placental tissue, 4 µm thick sections were obtained in a microtome (American Optical, Studio City, California, USA, Spencer model), subjected to standard staining with hematoxylin and eosin (H&E) in order to analyze the histopathological changes in an Olympus BX53 optical microscope with Olympus DP72 camera attached. Images were captured using Image-Pro Plus software version 7.0 (Media Cybermetics, Carlsbad, CA, USA).

### 2.4. Ultrastructural Analysis

Placental tissue samples were prefixed in 2.5% glutaraldehyde in 0.1 M cacodylate buffer pH 7.4 and postfixed with 1% osmium tetroxide. Dehydration was performed from a graded series of acetone solutions (30 to 100%) before infiltration into increasing baths of Epon resin (3:1, 1:1, and 1:3 acetone/Epon) and inclusion in Epon at 60 °C for 72 h. Ultrathin sections of ~60 nm were cut in an ultramicrotome (Zeiss) and contrasted with uranyl acetate and lead citrate for analysis of cellular changes in a transmission electron microscope (Hitachi HT 7800), as well as viral particle detection.

### 2.5. Immunoperoxidase Reaction

Initially, the endogenous peroxidase activity was blocked in the sections using 3% hydrogen peroxide for 15 min, and the sections were washed with phosphate-buffered saline (PBS) 3 times, for 5 min each time. The slides were submitted to antigen retrieval by citrate buffer, pH 6.0, for 20 min at 60 °C. Then, the sections were washed again with PBS and unspecific antibody labels blocked by incubation in PBS/BSA (bovine serum albumin) 3% for 20 min at RT. The sections were then incubated with the primary antibody (produced in house) diluted in PBS (1:1500) in a humid chamber overnight at 4 °C The production of this antibody protocol, with the immunizations, its titration, and other characteristics were described in detail previously [36]. Briefly, horses were immunized with trimeric spike glycoprotein (Residues 1-1208) in the prefusion conformation for production of hyperimmune globulins against SARS-CoV-2. The next day, the sections were washed with PBS and incubated with biotinylated secondary antibody (Biogen, Spring, Cambridge, MA, USA) for 1 h and subsequently with streptavidin (Biogen, Spring) for 30 min at room temperature. After washings with PBS, the products of the immunoreaction were visualized using the substrate diaminobenzidine (DAB) (Biogen, Spring) and counterstained with hematoxylin. The slides were finally dehydrated in increasing concentrations of alcohol, 70, 90, and 100%, and xylol and mounted with entellan and coverslips for further observation under an optical microscope.

## 3. Presentation of Cases

### 3.1. Cases Description

Clinical and demographic description are described in the table below (Table 1):

### 3.2. Macroscopic Evaluation of SARS-CoV-2 Infected Placentas

All collected placentas were analyzed for macroscopic characteristics, but only one control and two infected were photographed. In the macroscopic evaluation of the control placenta, it was possible to observe the placental disc with normal characteristic aspects, with a discoid shape, composed of the fetal face, covered by membranes (Figure 1A) and the opposite maternal face, divided into wine-colored and intact lobes (Figure 1B). After cleavage, it was possible to observe the spongy-looking wine tissue with a thickness of 2–3 cm (Figure 1C).

In Case 1, the placenta presented 21 × 19 cm, 523 g, oval in shape and regular edges, partially marginate. Fetal face of bluish/gray color, with discrete greenish areas covered by a transparent membrane and partially detached amnion, accentuated trabeculation, four dispersed and little congested vascular trunks. A wine-colored maternal face, well-defined and intact wolves. After cleavage, the parenchyma is pink/reddish and spongy with an average thickness of 2 cm (data not shown).

In Case 2, an aspect similar to the massive deposition of fibrin was observed, with light brown and dense areas along the basal decidua and permeating the spongy tissue, in an irregular way. In addition, diffuse winey areas were observed, sometimes outlining areas of recent infarctions (wine-red areas) and old ones (brown-white areas) (Figure 1D–F).

In Case 3, the placenta was measuring 18.5 × 16.5 cm and 383 g, with an oval shape and regular edges. Pink/bluish colored fetal face, covered by a hypotransparent membrane, moderate to severe trabeculation, three dispersed and congested vascular trunks. Wine-red maternal face, poorly delimited wolves, may be intact or frayed. After cleavage, the parenchyma had a wine-red and spongy color with an average thickness of 2.5 cm (Figure 1G,H).

In Case 4, the placenta was measuring 23 × 18 cm, 549.9 g, oval shape, intact, irregular edge. Fetal face of a blue-violet color covered by a transparent membrane, with partially detached amnion, light trabeculation, and four vascular trunks slightly congested, with dispersed distribution. Maternal face with wine-red wolves well defined and frayed in some areas. Presence of calcifications and adhered peripheral clots. After cleavage, the parenchyma showed a wine color, with a spongy consistency, with an average thickness of 1.5 cm. Presence of an area of peripheral infarction measuring 2 cm (data not shown).

In Case 5, the placenta was measuring 16 × 15.5 cm and 331 g, with an oval shape and regular edges. Fetal face with a bluish pink color, covered by a hypotransparent membrane, with accentuated trabeculation and four congested vascular trunks, with dispersed distribution. Presence of subchorionic fibrin deposits. Maternal face with wine-red wolves well defined and superficially frayed in some areas. After cleavage, the parenchyma showed a wine color, spongy consistency, and little evident lobar design, with an average thickness of 2 cm (data not shown).

### 3.3. Histopathological Changes in Placental Tissue Infected by SARS-CoV-2

The microscopic analysis was performed in order to investigate histopathological alterations in infected placenta. Noninfected control placenta tissue showed a regular aspect of the decidua and chorionic villi (Figure 2A). Fibrin deposition was noted in Cases 1 (Figure 2B), 3 (Figure 2D), and 5 (Figure 2F). Inflammatory infiltrate was observed in: (i) chorionic villi, named as villitis was identified in Cases 2 (Figure 2C), 3 (Figure 2D), and 4 (Figure 2E), and (ii) intervillous space (intervillositis) in Case 5 (Figure 2F). In Case 1, there was necrosis of trophoblast cells (Figure 2B). Ultimately, in the decidua of Case 4, blood vessel thrombosis was noted (Figure 2E).

### 3.4. SARS-CoV-2 Spike Protein Detected in Placental Tissues

In order to investigate which cells were infected, immunohistochemistry was performed, using an anti-SARS-CoV-2 spike protein antibody. In the control placenta, there was no detection, as expected (Figure 3A). On the other hand, the infected placenta exhibited detection in different cell types. In the fetal portion of the placenta, the detection was in the (I) villous stroma in Case 1 (Figure 3B); (II) fetal cells (inside fetal capillaries) in Cases 1 (Figure 3C), 3 (Figure 3G), 4 (Figure 3K), and 5 (Figure 3M); in (III) trophoblast cells in Cases 2 (Figure 3E,F) and 4 (Figure 3I); and in (IV) Hofbauer cells in Case 4 (Figure 3L). In the maternal portion, spike protein was detected in maternal cells (in intervillous space) in Cases 3 (Figure 3H) and 4 (Figure 3I,J).

### 3.5. Placenta Ultrastructural Changes and Presence of Viral like Particle

In order to explore the placenta ultrastructural changes resulting from SARS-CoV-2 infection, we analyzed by transmission electron microscopy three of the infected samples (Cases 1–3), as well as a prepandemic control sample for comparison purposes. In the control, it was possible to observe syncytiotrophoblast cells with normal aspects, multinucleated, with a cell membrane rich in microvilli on the face facing the intervillous space and high production of secretion vesicles. The cytotrophoblast in the control has a normal appearance, with cytoplasm rich in organelles such as the endoplasmic reticulum and mitochondria (Figure 4A–C). In the analysis of the infected placentas, we noticed a very similar pattern of alterations between the samples, with predominant characteristics of apoptosis. Syncytiotrophoblasts are retracted, with pyknotic nuclei of condensed and peripheral chromatin, loss of microvilli, secretion vesicles, and especially the presence of a large amount of apoptotic bodies and myelin figures throughout the cytoplasmic space (Figure 4D,G–I,M–O). Cytotrophoblasts from infected samples are retracted, with nuclei also retracted and pyknotic, in addition to apoptotic bodies in the cytoplasm, absence of mitochondria, and endoplasmic reticulum with dilated cisterns (Figure 4E,F,J–L). In Cases 1 and 2, it was possible to observe the presence of viral particles of approximately 70 nm, compatible with the size of SARS-CoV-2 (Figure 4E,F,K,L).

## 4. Discussion

Pregnant women infected with SARS-CoV-2 are at increased risk for adverse pregnancy outcomes, including preterm delivery, poor fetal vascular perfusion, and premature membrane rupture [37,38,39]. Furthermore, disease severity has been strongly associated with the severity of pregnancy complications [38,40,41,42,43]. On the other hand, pregnant women with mild symptoms or asymptomatic had similar results to pregnant women not infected with SARS-CoV-2 [38,40].

In this study, we identified that the placenta of pregnant women infected with SARS-CoV-2, who evolved with mild symptoms of COVID-19, presented areas of infarction, fibrin deposits in the intervillous space and in the chorionic villi, as well as areas of calcification. Additionally, we observed inflammatory changes (intervillositis, villitis, and acute and chronic deciduitis) and maternal and fetal hypoperfusion changes, which can influence placental homeostasis, leading to complications during pregnancy. These findings are similar to those found in other studies involving pregnant women infected with SARS-CoV-2 [23,26,44].

The previously mentioned inflammatory lesions associated with infarct areas and fibrin deposition are associated with poor fetal vascular perfusion and poor maternal vascular perfusion [44,45]. Several studies have pointed out the high incidence of poor maternal vascular perfusion in pregnant women infected with SARS-CoV-2 [25,43,46,47,48,49,50,51,52,53,54,55]; therefore, it can be said that this condition is closely related to viral infection and inflammation. Poor maternal vascular perfusion is a pattern of injury associated with decreased vascular supply and is associated with clinical disorders such as fetal vascular thrombosis, abnormal umbilical cord insertion, umbilical cord hypercoil, and maternal hypercoagulable state, which can result in fetal growth restriction and even premature delivery [56].

In the ultrastructural analysis of the placentas, it was possible to observe alterations that suggest an intense apoptotic process of the local cells. This process of apoptosis has been observed in other cell types infected with SARS-CoV-2 [46,57]; thus, the occurrence of apoptosis observed may be related to infection by the virus and can be further studied. In addition, we detected the presence of “virus-like particles”, consistent with the dimensions of SARS-CoV-2 [2,4,47,58,59]. Other studies have already detected SARS-CoV-2 in syncytiotrophoblasts, fibroblasts, microvilli, and fetal endothelial capillary cells close to the villus surfaces, as well as in intravascular mononuclear cells [56,60], corroborating the hypothesis that the “virus-like particles” found in our work are SARS-CoV-2 particles. The presence of the viral particles in the tissue so long after the period of symptoms suggests persistence of the virus that may be associated with long-lasting COVID-19 (or chronic COVID syndrome).

In line with the previous result, we detected the spike protein both in the fetal portion of the placenta (villous stroma, within fetal capillaries, trophoblastic cells, and Hofbauer cells) and in the maternal portion (intervillous space). In this study, we detected the presence of viral antigens in trophoblastic cells in the villi, which suggests that the virus can infect these cells that make up the placental barrier. Thus, virus entry could occur not only with the aid of the TMPRSS2 coreceptor, but also CD147 [16]. The detection of the S protein, together with the presence of these viral particles, corroborates the positive result in these placentas from the real-time RT-PCR, which proves the persistence of the virus even many weeks after the symptoms and suggests that the viral infection contributed to the appearance of lesions in the placental tissue. Viral persistence was reported by other authors, including a case of an asymptomatic mother, with viral persistence in the placenta and proven transplacental transmission once SARS-CoV-2, was detected in the amniotic fluid and fetal membranes [61,62]. Macrophages were suggested as a possible site of persistence of SARS-CoV-2; however, it is not fully understood [61]. More studies are needed to elucidate the mechanisms of placental persistence, as this organ may be a viral sanctuary.

Furthermore, in our study, one of the newborns presented anti-SARS-CoV-2 antibodies, corroborating the passage of maternal immunity to the baby, as observed in previous studies, or suggesting that vertical transmission may occur, which should be further studied [17,29,38,63]. Other work has already shown that vertical transmission of COVID-19 can occur in the third trimester, and the rate is approximately 3.2% [64].

The investigation of this work on the macroscopic, histopathological, and ultrastructural changes found in the placentas, as well as viral detection, may contribute to a better understanding of the disease on vertical transmission and its possible effects on fetal development.

## 5. Conclusions

It was observed that the virus was able to reach the placenta from the positive result in immunohistochemistry by peroxidase in all samples and in some also by RT-qPCR. The presence of “ virus-like particles” in placental cells with a size compatible with that of SARS-CoV-2 confirms the presence and viral replication in cells of this tissue and suggests persistence of the virus that may be associated with long-lasting COVID-19. Macroscopic, histopathological (inflammatory and hypoperfusion), and ultrastructural (apoptosis-related) changes were found in the infected placentas. The identification of these alterations contributes to a better understanding of the pathogenesis of the infection in the maternal–fetal context.

## Figures and Tables

**Figure 1 viruses-14-01885-f001:**
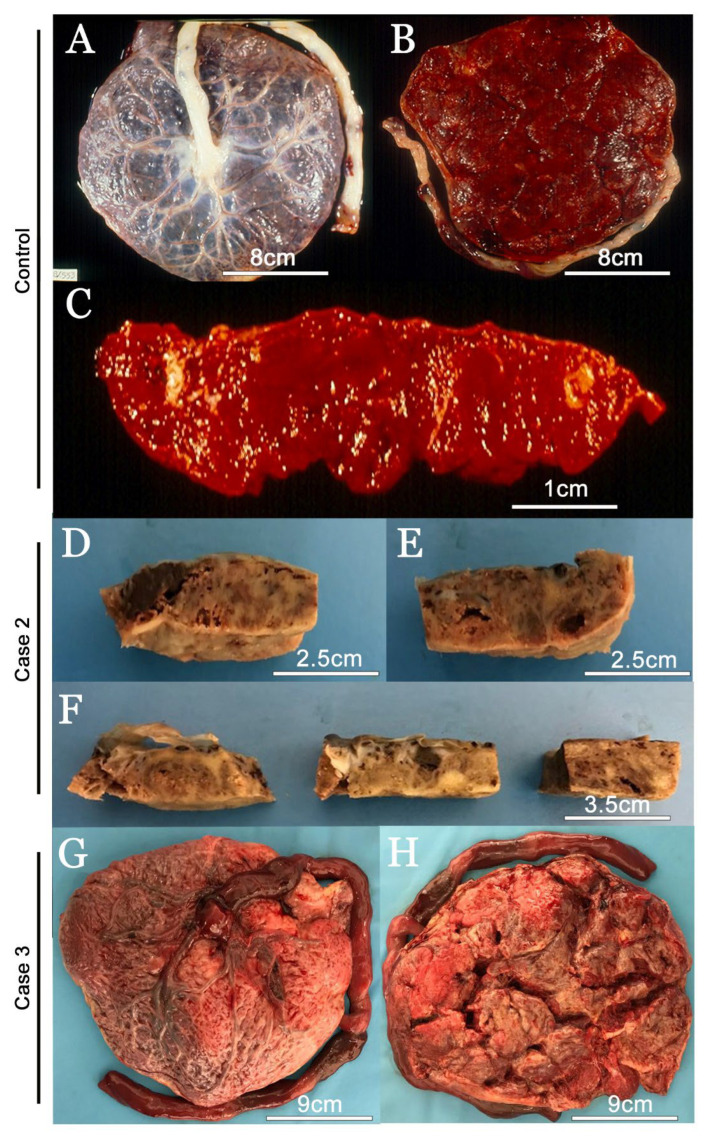
Macroscopic evaluation of placentas. (**A**–**C**) Prepandemic COVID-19 control fresh placenta; (**A**) fetal face; (**B**) maternal face; (**C**) cleaved placenta; (**D**–**H**) SARS-CoV-2-infected placentas; (**D**–**F**) cleaved and fixed placenta of the second case; (**G**) fresh fetal face of the third case; (**H**) fresh maternal face of the third case. These are photographs of representative placenta from participants who had pregnancies during the COVID-19 pandemic.

**Figure 2 viruses-14-01885-f002:**
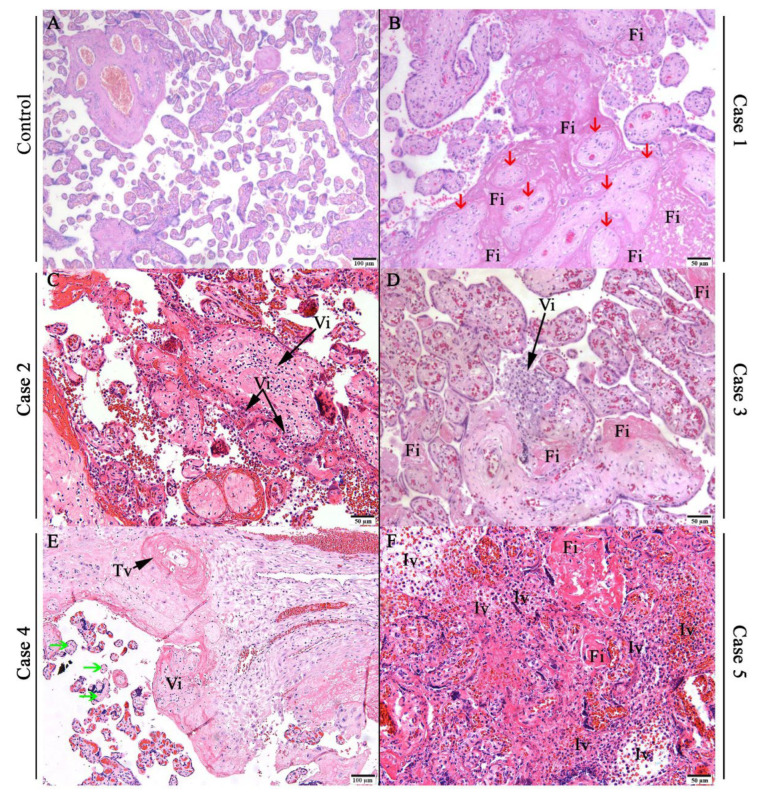
Histopathological changes in SARS-CoV-2-infected placentas. (**A**) Prepandemic control placenta with regular aspect. (**B**–**F**) Representative microphotographs of SARS-CoV-2-infected placentas. Fibrin deposition (Fi); trophoblastic necrosis (red arrow); villitis (Vi); chronic villitis (Vi); avascular villi (green arrow); decidual vessel thrombosis (Tv); and intervillitis (Iv).

**Figure 3 viruses-14-01885-f003:**
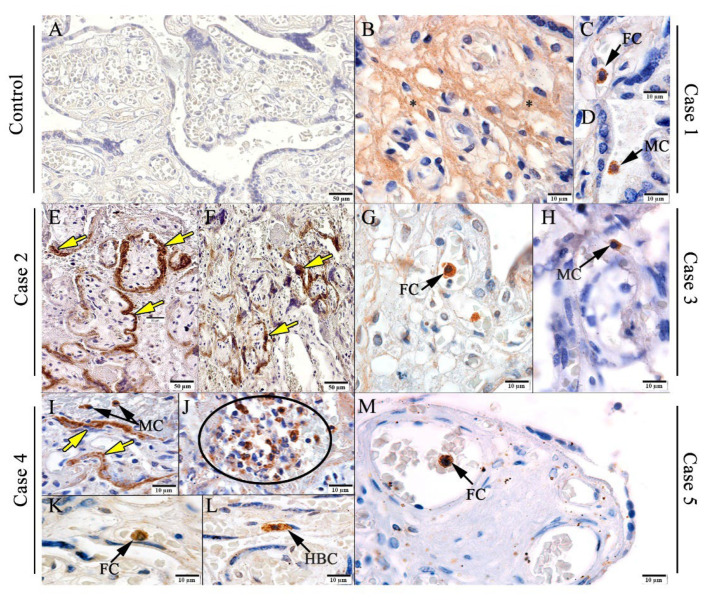
Detection of spike protein in placental cells. (**A**) Pre-pandemic control placenta. (**B**–**M**) Spike protein detected in: (**B**–**D**) Representative microphotographs of SARS-CoV-2 infected placentas. HBC—Hofbauer cells; FC—circulating cells of fetal capillary; yellow arrows—trophoblastic cells; MC—macrophages.

**Figure 4 viruses-14-01885-f004:**
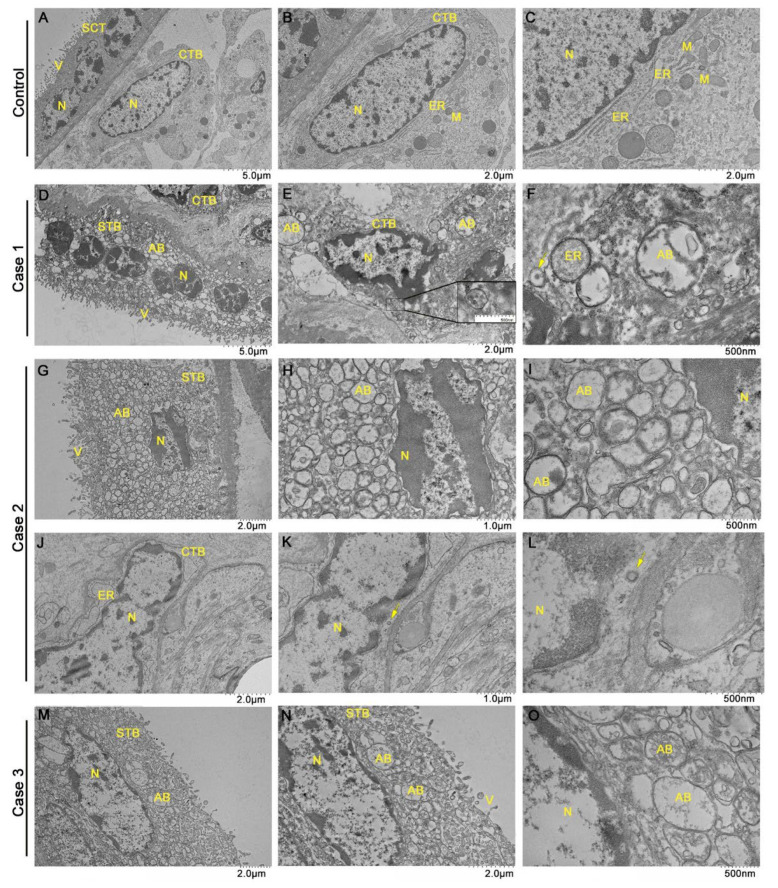
Ultrastructural changes in SARS-CoV-2 infected placental cells and presence of viral particles. (**A**–**C**) Prepandemic control placental sample with normal-looking syncytiotrophoblast (STB) and cytotrophoblast (CTB) cells. (**D**–**O**) Samples from infected placentas, with altered morphology. (**E**) Higher magnification of the viral cluster area. (N) nucleus; (M) mitochondria; (ER) endoplasmic reticulum; (V) villi; (AB) apoptotic bodies. Arrowheads indicate viral particles of approximately 70 nm, compatible with the size of SARS-CoV-2.

**Table 1 viruses-14-01885-t001:** Clinical and demographic description of the five cases. * This case was considered for analysis because, although asymptomatic, the baby was stillborn during the high pandemic outbreak period and subsequently confirmed the SARS-CoV-2 detection by immunohistochemistry.

	Case 1	Case 2	Case 3 *	Case 4	Case 5
**Age (year-old)**	37	38	39	28	27
**Symptoms**	▪ Fever ▪ Headache ▪ Dry cough ▪ Nasal congestion ▪ Runny nose ▪ Myalgia ▪ Fatigue ▪ Nasal bleeding	▪ Fever ▪ Cough	Not presented symptoms	▪ Chills ▪ Headache ▪ Dry cough ▪ Nasal congestion ▪ Rhinorrhea ▪ Myalgia ▪ Arthralgia ▪ Fatigue ▪ Vomiting ▪ Nausea	▪ Fever ▪ Chills ▪ headache ▪ Dry cough ▪ Sore throat ▪ Sneezing ▪ Nasal congestion ▪ Rhinorrhea ▪ Anosmia ▪ Loss of taste ▪ Chest pain ▪ Shortness of breath ▪ Myalgia ▪ Arthralgia ▪ Fatigue ▪ Abdominal pain ▪ Diarrhea
**Onset of symptoms** **(period of gestation)**	19 weeks	38 weeks	-	39 weeks	35 weeks
**Delivery**	39 weeks and 4 days	38 weeks	39 weeks	41 weeks and 1 day	37 weeks and 4 days
**SARS-CoV-2 diagnosis**	Nasopharyngeal PCR was not performed. IgG	No nasopharyngeal PCR or serology was performed	Negative nasopharyngeal PCR	IgG and IgA in the day of delivery	IgG in the day of delivery
**P** **re-existing disease**	No	No	Asthma	Obesity and hypothyroidism	Asthma
**Newborn information**	Apgar 9/10	Fetal death	-	Apgar 9/10, presence of anti-SARS-CoV-2 IgG antibodies	Apgar 7/8
**SARS-CoV-2 RT-PCRq of the placenta sample**	Positive	Positive	Not performed	Not performed	Positive

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
