# Peer review of "SARS-CoV-2 Is Persistent in Placenta and Causes Macroscopic, Histopathological, and Ultrastructural Changes"

_viruses, 2022, doi:10.3390/v14091885_

Round 1

Reviewer 1 Report

André Luiz N. Parcia et al present a series of 5 cases of pregnant women with COVID-19. Considering that the type of this manuscript is a case report, I consider that the presentation is wrong and does not comply with the CARE guidelines. Therefore I am of the opinion that this manuscript needs several major changes, as follows.

1. Introduction

The authors should make clear their views on the relevance of the reporting of these cases to the scientific community.

2. The title of Chapter 3 should be changed from "Results" to "Presentation of Cases". I believe that there can be no results without statistical analysis. After all, the authors only present the findings of each individual case.

3. Therefore, I am of the opinion that each case should be presented individually, presenting the patient's data, clinical findings, diagnostic data, microscopic evaluation of the placenta, a chronology of the disease, the interventions used and the results. I believe that the current state of presentation is confusing to the reader, and such a presentation could be moved to the discussion section. I believe that comparison of histopathological findings is imperative to be included in the discussion.

4. Present more detailed information about: patients (De-identified patient specific information, symptoms, personal history, previous relevant interventions), clinical findings (physical examination), therapeutic interventions, outcomes.

5. The discussion chapter should include presentation and comparison of histopathological results (as you have written in the results section of the current form of the manuscript). As I said before I think the presentation in chapter 3 would fit better in the discussion section. Please also discuss the findings according to intervention (treatment performed), symptomatology, severity of disease.

In conclusion, I consider that this manuscript is not publishable in its current form, and kindly invite the authors to report these cases according to CARE guidelines (https://www.care-statement.org/checklist).

Author Response

Dear Reviewer, 

We are very grateful for your review, with the modifications, there has been an improvement in our work. Here is our point-by-point response:

1- We highlitghted to the introduction some points that show the relevance of the research (Lines76-79 and 89-92). This work was carried out with rare material from Rio de Janeiro, at the peak of the pandemic before vaccination. Therefore, the description of these cases can contribute to a better understanding of the pathogenesis of the disease in pregnant women.

2- Our paper was originally designed to be an original, since comparisons with control placentas in different techniques were used. However, we understand that because it is a mostly descriptive work, we accept the editor's suggestion to proceed as a case report. Therefore, we accepted the suggestion and changed the topic "Results" to "Presentation of Cases" (Line 162).

3-  We accept all of the referee's suggestions, but on this one, we think that the best way to expose the histopathological results is to compare them with the control in the "Presentation of Cases" session and discuss them in the "discussion" session. We kept it this way also because the two other referees approved this model. We have added more information to the discussion.

4- We have added a table to the article in order to better explain the clinical data and patient demographics, as was also suggested by another referee. The table has the symptoms, the tests performed, the personal history of the patients, as was suggested.

5- As suggested by the reviewer, we have moved the paper to the case report template, added points suggested by all reviewers to make it better fit the suggested template. Thank you for your considerations about the manuscript. We highly appreciated the comments and opinion.

Reviewer 2 Report

In such a way that I consider that the work can be published, and some recommendations to enrich this work could be considered.

1.- Within the introduction they mention the presence of another corrector and the possibility that cellular components of the placenta express it, however in the discussion they do not return to this and above all the relevance that it could have when significant tissue damage is observed , and as they comment that its distribution is in the basement membrane of the trophoblast, it would now be possible for the virus to access this receptor.

2.- An interesting piece of information in their work is the hypothesis that they handle about a viral persistence given that their diagnosis was weeks prior to delivery; and the findings are interesting to identify viral RNA, spike protein and virus-like particles in the tissue. Could you also delve into this, that is, according to the literature in the general population, is there persistence of the virus? Is a mechanism for this already known? A hypothesis would be interesting.

Author Response

Dear Reviewer, 

We are very grateful for your review, with the modifications, there has been an improvement in our work. Here is our point-by-point response:

1-  The CD147 may be an alternative co-receptor to favor virus entry, and is found in trophoblastic cells.  Therefore, we added this information also in the discussion (lines 349-352). 

2- In fact, this is one of the most important findings of this work and little is known about persistence in the literature, since this virus has recently emerged. We have added to the discussion a paper that discusses the role of macrophages in persistence, but it is still poorly understood (lines 359-360).  We suggest in the paper that more studies be done focused on this to further clarify if the placenta can be a viral sanctuary (Lines 356-360). Thank you for your considerations about the manuscript. We highly appreciated the comments and opinion.

Reviewer 3 Report

The manuscript entitled “SARS-CoV-2 is persistent in placenta and causes macroscopic, histopathological and ultrastructural changes” by Parcial and Salomão et. al. is an interesting and well-written manuscript. In this manuscript, authors showed the SARS-CoV-2 infection and post-infection changes in the placental tissue by using qRT-PCR, histopathological analysis, and immunoperoxidase assays. Authors were also able to show the presence of ‘virus-like particles’ in the infected samples using transmission electron microscopy. Authors have performed the study in a very systematic manner with lots of available supportive information. However, there are some major concerns related to this study, and still has some room for the improvement. I have the following comments that need to be addressed before the manuscript will be accepted for the publication;

  1. Now a days, multiplex qRT-PCR assays are mostly used for the accurate diagnosis of SARS-CoV-2. If the result is positive for all the gene sets, then only, in that case, the specimens will be considered positive, otherwise not. In this study, authors need to use at least two to three sets of primer-probe pairs for their qRT-PCR analysis in order to specifically detect the SARS-CoV-2. So, authors need to find out an additional target region (specific for SARS-CoV-2) for their qRT-PCR analysis because targeting the E gene is specific for lineage B-betacoronavirus.
  1. Why authors didn’t perform the qRT-PCR for nasopharyngeal swabs, which is the first COVID-19 test that was needed to perform for the patient admitted? Also, the authors performed qRT-PCR for only the placental tissue, not the other related specimens. It would be great if the authors can show the infection in different maternal and neonatal specimens and demonstrate the transplacental transmission of SARS-CoV-2 infection by using qRT-PCR and determine the viral load for each sample (for ex; can be expressed as Log copies/million of cells for the placenta).
  1. Previous studies have employed a variety of techniques to diagnose placental infection, including qRT-PCR, immunohistochemistry, RNA in situ hybridization (RNA-ISH), and electron microscopy, and were able to clearly show the virus-like particles in the placentas. So, it would be great if the authors can compare their findings with the previously published reports and point out the significance of this study and why this study is different from others. It would be great to demonstrate the transplacental transmission of SARS-CoV-2 infection through the placenta and to discuss the possibility and frequency of vertical transmission. It would provide strength and novelty to the manuscript.

  1. Summarize the clinical and demographic description of the patients, qRT-PCR findings, histopathological studies, and macroscopic analysis performed in a tabular form, which will be easy to better understand.

Author Response

Dear reviewer. 

We are very grateful for your review, with the modifications, there has been an improvement in our work. Here is our point-by-point response:

1-  We would like to add new analyses for other genes, however, here in Brazil ordering primers takes a long time in delivery and now in the pandemic period these logistics are even more complicated. Besides the detection of viral RNA, we perform in all sample’s immunohistochemistry for the detection of viral antigen and also the detection of virus-like particles by transmission electron microscopy.

2-  The patients from whom we collected these samples were admitted to hospitals in 2020, at the beginning of the pandemic period, in which most public hospitals in our country did not yet have PCR testing available for the analysis of nasopharyngeal samples. Therefore, this test was not performed. For the analyses, we depended on collaborations with doctors and nurses from the hospitals, and other samples were not collected. We were able to perform it on the placentas, because the samples were properly frozen for later analysis. Since the PCR tests were still performed in the hospitals collaboratively, for diagnostic purposes it was considered only positive or negative and we did not get more precise information such as the exact viral load.

3-   We correlated our findings with those in the literature and added the percentage of vertical transmission already reported (line 365). The cases evaluated are from the pre-vaccination period and the viral persistence shows that the placenta can be a sanctuary for the virus, which was shown for the first time.

4-  We kept the histopathological findings as the figure description, as suggested by another reviewer. However, we arranged the clinical data, demographic description of the patients, qRTPCR findings in table form, as suggested by the reviewer.Thank you for your considerations about the manuscript. We highly appreciated the comments and opinion. 

Round 2

Reviewer 3 Report

The authors tried to address most of the comments raised in this study except by performing some additional suggested experiments, which can provide further strength to the manuscript. Some of the suggested modifications have been taken into account in improving the quality of the article. The revised version of the manuscript is clear, concise, and well-written. However, there is still some room for improvement that need to be addressed before the manuscript will be accepted for the publication;

1)     Please add scale bars in all the images in figure 1. Also, please increase the font size of the scale bar in figure 2, Figure 3, and figure 4, so that they can be easily visualized.

2)     Please use the same format of writing SARS-CoV-2 throughout the manuscript. For example;

Line 122: Sars-CoV-2 ----à SARS-CoV-2

Line 172: SARS-COV-2 ------à SARS-CoV-2

Line 237: SARs-CoV-2 ---à SARS-CoV-2

3)     Correct the following sentences, wherever applicable throughout the manuscript;

Line 135; Hematoxylin and Eosin (HE) ------à Hematoxylin and Eosin (H&E)

Line 377; RT-PCRq ---à RT-qPCR

Author Response

Dear reviewer,

All suggested points of modifications to the text and figures have been made. We appreciate the review and suggested improvements.

Best regards.